# Study of Surface Integrity of Titanium Alloy (TC4) by Belt Grinding to Achieve the Same Surface Roughness Range

**DOI:** 10.3390/mi13111950

**Published:** 2022-11-11

**Authors:** Guiyun Jiang, Zeyong Zhao, Guijian Xiao, Shaochuan Li, Benqiang Chen, Xiaoqin Zhuo, Jie Zhang

**Affiliations:** 1College of Mechanical and Vehicle Engineering, Chongqing University, No. 174, Shazhengjie, Shapingba, Chongqing 400444, China; 2State Key Laboratory of Mechanical Transmissions, Chongqing University, No. 174, Shazhengjie, Shapingba, Chongqing 400044, China; 3College of Mechanical and Electrical Engineering, Nanjing University of Aeronautics & Astronautics, Nanjing 210016, China

**Keywords:** belt grinding, Ti-6Al-4V, surface roughness, surface integrity

## Abstract

Titanium alloy materials are used in a variety of engineering applications in the aerospace, aircraft, electronics, and shipbuilding industries, and due to the continuous improvement of the contemporary age, surface integrity needs to be improved for engineering applications. Belt grinding parameters and levels directly affect the surface integrity of titanium alloys (TC4), which further affects the fatigue life of the titanium alloys during service. In order to investigate the surface integrity of titanium alloys at different roughness levels, the surfaces were repeatedly ground with the same type and different models of abrasive belts. The results showed that at roughness Ra levels of 0.4 μm to 0.2 μm, the compressive residual stresses decreased with increasing linear velocity and there were problems with large surface morphological defects. At the roughness Ra of 0.2 μm or less, grinding improves the surface morphology, the compressive residual stress increases with increasing feed rate, and the surface hardness decreases with increasing linear velocity. In addition, the research facilitates the engineering of grinding parameters and levels that affect surface integrity under different roughness conditions, providing a theoretical basis and practical reference.

## 1. Introduction

Titanium alloys are widely used in various engineering applications in the aerospace, aircraft, electronics, and shipbuilding industries [1] due to their excellent overall properties, such as low density, high strength-to-weight ratio, and excellent corrosion resistance [2,3]. Due to the continuous improvement of modern properties, titanium alloy materials require continuous improvement of surface integrity to achieve serviceability [4,5,6]. However, it is well known that surface integrity is an important area of aero-engine research, as surface integrity determines the performance of the engine and, in turn, the proper operation of the entire system. Aerospace often uses abrasive belts to grind blades to improve surface integrity [7,8] and thus fatigue life. Fatigue life is mainly influenced by roughness [9], hardness [9,10], and compressive residual stress [11]; therefore, fatigue life can be improved by improving grinding parameters and levels for higher roughness, increased surface hardness, and compressive residual stress [12].

Many studies have found that the surface roughness after grinding has a great influence on fatigue life [9]. Surface micro-scratches, as one of the stress concentration factors [13], cannot be neglected in fatigue analysis and fatigue life assessment of metallic materials, especially in high cycle fatigue and ultra-high cycle fatigue [14]. Surface micro-scratches are inevitably caused during the grinding process and are considered “fatigue defects”. These defects can cause stress concentrations that weaken the material’s resistance to fatigue failure, resulting in cracking defects that can lead to reduced life. Zhu et al. [15] studied the effect of abrasive belt grinding force on roughness, which was further divided into grinding linear speed and grinding depth, and the grinding linear speed and grinding depth resulted in different roughness. Zhao et al. [16] found that surface roughness has a great influence on contact noise, vibration, friction, and wear of work, and surface damage can be eliminated by improving roughness through grinding. Sun et al. [17] studied the roughness of silicon nitride ceramics with different grinding parameters and found that the linear speed of grinding affected the roughness the most, and the surface roughness increased as the radial feed rate increased. Tan et al. [18] found that the reduction of grinding roughness from Ra 0.43 to 0.15 μm for titanium alloy had a greater effect on fatigue life than the reduction of residual compressive stress from 600 MPa to 300 Mpa. When the surface roughness is less than a certain threshold (Ra 0.5–0.1 μm), the change in roughness no longer affects the fatigue strength [19]. Researchers have also found significant changes in the surface morphology of the material when the surface roughness produced by grinding is based on a threshold value of 0.2 μm [19,20]. The difference in material removal rate and thus roughness due to processing parameters was classified into 0.4 μm and 0.2 μm based on the division of industries and the effect of equipment removal rate [21,22,23]. Based on the above studies, most researchers just concentrate on grinding to reduce the surface roughness and do not care about the effect of surface roughness on other surface integrities under the condition of lower surface roughness.

Numerous studies in recent years have shown that the use of abrasive belt grinding improves surface integrity, not only in terms of surface roughness but also in terms of surface hardness and residual stresses [10,24]. Grinding generates a lot of heat, which makes the material surface hard and can lead to an increase in hardness. Gao et al. [25] increased the surface hardness by studying that grinding generates a lot of heat and therefore transforms into a martensitic structure in the surface layer of the material, which reduces the softening width by double-pass grinding. Liu et al. [26] studied the surface hardening produced after grinding, and the results showed that the surface hardness and thickness of the hardened layer were mainly affected by the grinding depth, and the grinding depth and rotational speed should be controlled to obtain the maximum surface hardness and thickness of the hardened layer. Xiu et al. [27] controlled the hardness by controlling the grinding heat and mechanical force, and thus the wear resistance of the workpiece. Nowadays, most scholars and experts discuss fatigue life through residual stresses and increase residual stresses to achieve increased fatigue life [28,29,30]. Yao et al. [31] studied the fatigue strength of ultra-high strength steel in large structures in terms of residual stresses and found that different grinding wheel speeds resulted in different residual stresses and affected layers. Zhao et al. [32] investigated the effect of crystal orientation (CO) on the distribution of residual stresses induced by micro-grinding, and the results showed that the effect of CO on residual stresses was significant. Li et al. [33] studied the effect of residual stresses on the life of cams during grinding and finally found that temperature and grinding speed played a key role in the life of cams. Zhao et al. [11] studied the residual stress state on the surface of abrasive belt grinding tracks, for which the effect of different grinding process parameters on residual stresses was investigated, and the results showed that tensile residual stresses were found in the grinding direction, while residual stresses in the radial direction were mainly maintained in the form of compressive stresses. Shen et al. [34] studied the residual stress distribution in martensitic effective rigid after grinding, and the results showed that the residual stresses and the peak residual stresses were largely dependent on the grinding speed, with greater residual stresses occurring as the speed increased. B. Denkena et al. [35] investigated the residual stresses inside the tool after grinding, and larger grain size lead to more compressive residual stresses, while higher feed speed shifted the stresses to the tensile state. Chen et al. [36] investigated the residual stresses induced by ultra-high-speed grinding of difficult-to-machine materials, and the results showed that smaller grinding depths and higher grinding speeds are beneficial to achieving better residual stresses. Sun et al. [37] investigated the different residual stresses due to large changes in the heat gradient generated by grinding, and by optimizing the grinding process parameters, the residual tensile stresses could be effectively reduced and the residual compressive stresses increased. However, few papers have reported the differences between residual stresses, surface topography, and surface hardness at the same roughness, and which parameters and levels should be changed to increase residual stresses, surface topography, and surface hardness at the same roughness conditions.

With the above description, most researchers only study the effect of grinding on surface integrity and thus improve grinding parameters and level adjustment of surface integrity. The grinding parameters are the different parameters (step length, linear speed, feed rate, and grinding depth in this paper) and the grinding levels are the values for adjusting the parameters (1 mm, 2 mm, and 3 mm steps length in this paper). Most researchers have not investigated the effect on other surface integrity based on low roughness conditions. Therefore, based on the above description and the conditions of machine accuracy, this paper will study the effect of low roughness on other surface integrity under the conditions of surface roughness Ra below 0.4 μm, between 0.4 μm–0.2 μm, and in two ranges below 0.2 μm, respectively. In conclusion, this paper investigates the differences in residual stress, surface hardness, and surface topography of titanium alloys ground by different models of abrasive belts of the same type to the same roughness Ra (0.4 μm/0.2 μm) under different combinations of parameters and levels. From this, the parameters and levels that should be changed to improve the integrity of a single surface are derived from the experiments. The surface morphology of the titanium alloy surface was first examined, and the residual stress was measured. Next, the surface hardness was measured, and finally, the titanium alloy specimens were wire cut to observe the surface and subsurface morphology. In this paper, abrasive belt grinding of titanium alloy TC4 with different parameters and levels of roughness Ra between 0.4 μm–0.2 μm and below 0.2 μm were used to compare the different residual stresses, surface morphology, and surface hardness, which provides a basis for subsequent changes in single surface integrity.

## 2. Materials and Experimental Procedures

### 2.1. Materials

The material studied in this test is titanium alloy TC4, whose nominal chemical composition is commonly known as Ti-6Al-4V, the chemical composition is listed in Table 1, and the mechanical properties are listed in Table 2 [38]. The workpiece material was first manufactured and then cut into 100 mm × 50 mm × 4 mm sheets for the subsequent grinding test.

### 2.2. Experimental Procedure

#### 2.2.1. Surface Grinding

In this study, TC4 grinding was performed on a self-designed grinding machine with a maximum power of 12 W, a belt size of 1500 × 10 mm, a maximum workpiece feed rate of 2 m/min, and a maximum belt line speed of 30 m/s. The abrasive belts used were 3M’s 237AA pyramidal and nylon abrasive belts, made of a pyramidal mixture of alumina and resin, with particles as shown in Figure 1a, and the CK772T series from VSM was made of SiC stacked with particles as in Figure 1b.

Before each grinding test, grinding calibration was first performed using 237AA pyramidal abrasive belt A60#. The grinding depth during the calibration was 0.1 mm, the feed rate was 2 mm/s, and the grinding wheel linear speed was 15 m/s. After the calibration was completed, the grinding test started. Since the number of abrasive belt types did not correspond, it was not considered a variable and was used as a comparison test alone. The grinding test was designed with four variables of step, abrasive belt linear speed, feed rate, and downward pressure. Due to a large number of test parameters and levels, the orthogonal experiment was used to complete the test as shown in Table 3, in which four tests were designed. As shown in Figure 2, the step length in the table represents the distance between repeated grinding, the linear speed is the linear speed of the grinding wheel, the feed rate is the speed at which the plate moves relative to the grinding wheel, and the grinding depth is the depth at which the grinding wheel presses down on the plate. The grinding machine is listed in Figure 1c. The plate with a surface roughness Ra of 1.6 μm was first roughly ground until it reached a surface roughness Ra of 0.4 μm. Test I first ground the plate with 237AA pyramidal abrasive belt A100# to reach a surface roughness Ra of 0.5 μm or less, and then ground the surface roughness Ra to 0.4 μm or less with 237AA pyramidal abrasive belt A45#. Test 2 first ground the plate with CK772T abrasive belt P400# to reach a surface roughness Ra of 0.5 μm or less, next ground the surface roughness Ra to 0.4 μm or less with CK772T abrasive belt P600#. Ground with a surface roughness Ra of 0.2 μm or less, Test 3 first ground the plate with 237AA pyramid abrasive belt A100# to reach a surface roughness Ra of 0.5 μm or less, then the surface roughness Ra was ground to 0.4 μm or less 237AA pyramidal abrasive belt A45#. Finally, the surface roughness Ra was ground and polished to 0.2 μm or less using nylon abrasive belt. For Test 4, firstly, a CK772T abrasive belt P400# was used to grind the plate to reach a surface roughness Ra of 0.5 μm or less. Next, the CK772T abrasive belt P600# was used to grind the surface roughness Ra to 0.4 μm or less, then CK772T abrasive belt P800# was used to grind the surface roughness Ra to 0.3 μm. Finally, the CK772T abrasive belt P1200# was used to grind the surface roughness Ra to 0.2 μm or less. A titanium alloy plate was divided into 9 sample numbers for each of the 9 sample numbers of one test. The order of belt replacement for each sample number is listed in Figure 3. For each test, the grinding area was as shown in Figure 1d, and after the abrasive belt completely ground the surface, 5–7 repetitive measurements were taken using a portable roughness meter, and the average of 3 reasonable data was taken as the final data to achieve high reliability of the test results. In the fatigue study by Wang et al. [14], these micro-scratches were quantified in terms of surface roughness as two index values: Ra and Rz. Ra is widely used to analyze the effect of surface roughness on fatigue performance because of its advantages: first, Ra can reflect both the surface micro-geometry and the contour peaks of the surface morphology; second, it is simple for data collection and processing. Another roughness index value Rz is the sum of the highest and deepest groove depth at the smallest cross-section of the specimen. It is the critical value of surface roughness and cannot represent the surface quality. Therefore, in this paper, the surface roughness Ra is also used as a quantitative representation of surface micro-scratches roughness of abrasive belts during grinding, and several adjustments of parameters are made to achieve 0.4 μm/0.2 μm or less. In this paper, the roughness value Ra used is the arithmetic mean of the absolute Z(x)
(1)$$Ra=\frac{1}{l}\int_{0}^{l}|Z(x)|dx$$

The arithmetic means of the absolute value of the vertical coordinate Z(x) (the distance from each point on the measured profile to the reference line x) within the sampling length l.

#### 2.2.2. Surface Integrity Testing

Surface roughness measurement was performed after each grinding session using a hand-held roughness meter with a measurement length of 0.25 × 5 mm and a sampling length of 0.25 mm. 5–7 repeated measurements were performed each time, and the average of 3 reasonable data was taken as the final data to achieve high reliability of the test results. After each measurement, the grinding experiment continued until the end. After the grinding experiments, other parameters of the samples were started. First, the super depth-of-field measurements were performed using the VHX-1000C/VW-6000 super depth-of-field 3D microscope system designed and manufactured by KEYEN. The basic surface morphology can be obtained by taking pictures of the titanium alloy surface with 300–500 times magnification. Three places were selected for each sample number to be photographed. Since the burned part of the titanium alloy plate could not be accurately measured in the residual stress test, a judgment on the surface burn of the titanium alloy could be made in ultra-deep field photography. Next, a white light interferometer was used to scan the 3D profile to facilitate an in-depth understanding of the entire surface morphology. First, the special scanning lens was replaced with a magnification of 500×, and then a flat scanning position was found. For each sample number, three flat locations were selected for scanning, allowing a 3D view of the ground surface texture and the grooves created by grinding. Residual stresses were measured using a Proto XRD device, which required 15 min of warm-up to turn on. The equipment uses a Cu target with an inclination angle of −30 °C to 30 °C for the measurement and was irradiated 11 times. The first exposure is a single exposure, and after the single exposure data were finished, the backing was taken to filter the excess interference signal. After the single exposure data were adjusted, multiple exposures were performed. Five measurement positions were randomly selected for each area of the surface, and the average of 3 reasonable data was taken as the final result. Then, the surface hardness was measured, and in this paper, an in-situ nanohardness tester (Hysitron TriboView) was used to measure the surface hardness. Each measurement is performed in vacuo in order to prevent damage to the lens during the measurement. Then, the area where the hardness needs to be measured is positioned using the lens, and then the probe starts to press down. While the probe is pressing down, the device measures the precise measurement of the nanohardness, the experiment is selected to load a force of 4500–6500, and the array is measured 5 times, each time increasing by 500 N. The measurement results are 5 values, and 3 reasonable values are selected and averaged as the final result.

Finally, the sample was cut open and the specimen was first photographed with the super depth of field to find the electron microscope filming position. The electron microscope used was a Taseken scanning electron microscope, which mainly examined the surface morphology and sub-surface morphology of the specimen. The magnification of the electron microscope was 800–3500 times to have a more comprehensive understanding of the grinding affected surface and depth, and finally to observe the surface and sub-surface in combination with the super depth of field, white light 3D image, and electron microscope image.

## 3. Results

### 3.1. Effect of Grinding Parameters and Levels on Compression Residual Stresses

Table 4 is the surface roughness experimental results, and according to the table, the roughness degree and requirements can be observed. As Table 5 and Figure 4 show the graph of compression residual stress test results, according to the graph of test results, it can be seen that whether using a 3M abrasive belt or VSM abrasive belt, the compression residual stress between roughness Ra 0.4 μm–0.2 μm is higher compared to roughness Ra 0.2 μm or less, but the data are more fluctuating. Compared to the values of compression residual stress in test 1, test 2, and test 3, which were steadily increasing, the values of compression residual stress in test 4 were not regular. Next, the orthogonal experimental extreme difference analysis was performed, and the extreme difference analysis is listed in Figure 5. The extreme difference analysis experiment is intuitive and easy to understand, and the primary and secondary factors can be obtained by simple calculation and judgment. The R-value shown in the figure represents the extreme difference. In other words, the value was obtained by subtracting the minimum value from the maximum value of the resulting value [39]. It can be seen from the figure that the feed rate has the greatest influence, followed by the grinding wheel linear speed, step length, and finally grinding depth, which is subsequently analyzed in more detail according to this reference figure.

#### 3.1.1. Roughness Ra 0.4 μm–0.2 μm Compression Residual Stress Analysis

According to the extreme difference analysis in Figure 5, the factors that affect the compression residual stress the most are the grinding wheel linear speed and feed rate, so this paper will analyze the residual stress with the *Z*-axis as the compressive residual stress R value, with the grinding wheel linear speed and feed rate used as variables to analyze the compression residual stress, and the step length and the grinding depth used as variables to make another graph to further analyze the compression residual stress.

As shown in Figure 6, the compression residual stresses were first analyzed from the step length and the grinding depth. Figure 6a shows that the compression residual stresses decrease with the increase of steps in the step length of No. 2, No. 6, and No. 7 areas in test 1, and the compression residual stresses decrease by about 60%; the compression residual stresses decrease by 47% and can be analyzed from the areas of No. 1, No. 5, and No. 9. While analyzing the compression residual stress from the perspective of the grinding depth, analysis from the figure can be obtained—that the compression residual stress increases with the increase of the grinding depth, and the compression residual stress increases by 54% in the areas of No. 7, No. 8, and No. 9, and 42% in the areas of No. 4, No. 5, and No. 6. According to the analysis of test 3 in Figure 6b, the regularity of the step length and the grinding depth is the same as that of test 1. The largest increase in compression residual stresses affected by the step length is in the areas of No. 2, No. 6, and No. 7, which increased by 41%, and the average increase is 31%. In test 3, the compression residual stress increased by the grinding depth was less, and the maximum increase was 8%. According to the above analysis, it can be obtained that in the roughness Ra between 0.2 μm–0.4 μm, the step length affects the compression residual stress more than the grinding depth. If the compression residual stress is increased from the grinding depth and step length, the grinding depth can be increased or the step length can be reduced appropriately.

According to the analysis in Figure 6c, it can be obtained that the test 1 compression residual stress decreases with the increase of the linear speed, the area of No. 1, No. 6, and No. 8 decreased by 53%, while the compression residual stress in the area of No. 3, No. 5 and No. 7 decreased by 20%; it can be seen from the figure that the compression residual stress of the feed rate of 2 mm/s increased by 46% at maximum, compared with the feed rate of 4 mm/s, the compression residual stress for a feed rate of 1 mm/s is even greater, with a maximum increase of 19%. According to Figure 6d, the test 3 compression residual stresses decreased with the increase of the grinding wheel linear speed, and decreased by 25% in the areas of No. 1, No. 6, and No. 8; the feed rate increased by 27% at the maximum. According to the above analysis, it can be obtained that between roughness Ra 0.2 μm–0.4 μm, test 1 affects the compression residual stresses more than experiment 3 by the grinding belt’s linear speed. Compared with experiment 1, experiment 3 has a greater effect on the compression residual stress through the feed rate. If it is necessary to increase the compression residual stress, the grinding wheel linear speed can be reduced, or the feed rate can be set to around 2 mm/s.

#### 3.1.2. Compression Residual Stress Analysis for the Roughness Ra of 0.2 μm or Less

As shown in Figure 7, firstly, the effect of the step length and the linear speed on the compression residual stress of experiment 2 was analyzed, as shown in Figure 7a, and the compression residual stress decreased with the increase of the step length in the areas of No. 1, No. 5, and No. 9 (shown in Figure 4), which decreased by about 13%; the compression residual stress decreased by 14%, which can be analyzed from the areas of No. 3, No. 4, and No. 8. Next, the compression residual stress was analyzed from the point of view of the grinding depth, and from the analysis in Figure 7a it can be obtained that the compression residual stress increases with the increase of the grinding depth, and the compression residual stress increases by 29% in the areas of No. 4, No. 5, and No. 6. According to the analysis of test 4 in Figure 7b, the regularity of the effect of the step length, and the grinding depth, the compression residual stress is the same as that of test 2. The compression residual stresses affected by the step length increased the most in the areas of No. 2, No. 6, and No. 7, which increased by 46%. Figure 7b shows that the effect of the grinding depth on the compression residual stress is not very regular. According to the above analysis, it can be obtained that in the roughness Ra below 0.2 μm, the influence of step length on compression residual stress is larger than that of the grinding depth, and if it is necessary to increase the compression residual stress, the grinding depth can be increased or the step length can be decreased appropriately.

According to Figure 7c, analysis can be obtained. In Test 2, with the increase in linear speed compression, residual stress decreased, such as Figure 4, No. 1, No. 6, and No. 8 reduced by 9%, while No. 2, No. 4, and No. 9 area compression residual stress decreased by 7%; from Figure 7c we can see that the feed rate is not a regular maximum increase of 13%. Figure 7d shows that the test 4 compression residual stress decreases with the increase of grinding wheel linear speed, and No. 1, No. 6, and No. 8 decreased by 29%: the compression residual stress increases with the increase of feed rate by 48% maximum. Based on the above analysis, it can be obtained that below the roughness Ra of 0.2 μm, the grinding of test 2 has little effect on the compression residual stress, while the grinding of test 4 significantly affects the compression residual stress through the feed rate. If it is necessary to increase the compression residual stress, the grinding wheel linear speed can be reduced appropriately, or the feed rate can be set to an intermediate value.

### 3.2. The Effect of Grinding on Surface Hardness

Next, the surface hardness of the grinding surface was measured using the in-situ nano hardness tester, as shown in Table 6 and Figure 8, which shows the experimental result graph of the surface hardness. According to the experimental result graph, it can be seen that the data of the surface hardness are relatively stable, whether using 3M abrasive belts or VSM abrasive belts, and area 1 in test 4 is higher. According to the extreme differences, analysis data can be obtained in Figure 9 of the extreme difference analysis graph, and it can be seen that the feed rate and the abrasive belt linear speed have the greatest influence, then the step length and the linear speed, with subsequent analysis according to this reference graph.

#### 3.2.1. Surface Hardness Analysis for Roughness Ra 0.4 μm–0.2 μm

According to Figure 9, the factor that affects the surface hardness most is the grinding wheel linear speed and feed rate. Therefore, in this paper, the *Z* axis is used as the surface hardness R value, wheel linear speed and feed rate are variables, and the step length and the grinding depth are variables to make another regular graph to analyze the surface hardness.

From the analysis of the surface hardness from the step length and the grinding depth in Figure 10a, it can be seen that test 1 as a whole has the highest hardness affected by a step length of 2 mm and the lowest by a step length of 1 mm. Analysis of the step length in the areas of No. 2, No. 6, and No. 7 shows that the surface hardness increased by 21%; from the areas of No. 1, No. 5, and No. 9 it can be seen that the surface hardness increased by 33%. Next, the surface hardness was analyzed from the point of view of the grinding depth, and from the analysis of the figure it can be obtained that the surface hardness decreased with the increase of the grinding depth, and the surface hardness decreased by 20% in the areas of No. 1, No. 2, and No. 3. Analyzing the effect of test 3 on the surface hardness according to Figure 10b, the analysis from the area of No. 1, No. 5, and No. 9 obtained that the surface hardness increased by 36% with the increase of the step length; in the area of No. 3, No. 4, and No. 8, the surface hardness increased by 27%. From Figure 10b, it can be obtained that the surface hardness decreases with the grinding depth, and the greatest decrease is observed in areas No. 1, No. 4, and No. 7, with a decrease of 42%. The average decrease is 33%. According to the above analysis, it can be obtained that between roughness Ra 0.2 μm–0.4 μm, the effect of step length and grinding depth on surface hardness can be obtained. If the surface hardness needs to be increased, the step length should be set to about 2 mm and the grinding depth should be reduced.

According to the analysis of Figure 10c, it can be obtained that the surface hardness slightly decreases with the increase of the linear speed in test 1, e.g., the area of No. 3, No. 5, and No. 7 in the figure decreases by 5%, while the area of No. 2, No. 4, and No. 9 in the figure decreases by 5%; Figure 10c shows that the surface hardness slowly decreases with the increase of the feed rate, e.g., the area of No. 1, No. 4, and No. 7 decreases by 7.5%, and the area of No. 3, No. 6, and No. 9 decreases by 14%. According to Figure 10d, the surface hardness of test 3 decreases with the increase of grinding wheel linear speed, such as 34% in the areas of No. 2, No. 4, and No. 9, and 7% in the areas of No. 3, No. 5, and No. 7; the surface hardness increases with the increase of feed rates, such as 37% in the areas of No. 1, No. 4, and No. 7, and 32% in the areas of No. 2, No. 5, and No. 8. According to the above analysis, it can be obtained that between roughness Ra 0.2 μm–0.4 μm, test 3 grinding wheel linear speed and feed rate affect the surface hardness more, while test 1 has less effect on the surface hardness. If you need to increase the surface hardness, you can use VSM grinding belt to reduce the linear speed or increase the feed rate appropriately.

#### 3.2.2. Surface Hardness Analysis for Roughness Ra below 0.2 μm

From the analysis of the surface hardness of test 2 in terms of step length and grinding depth in Figure 11a, it can be seen that the surface hardness decreases with the increase of step for the step length in the areas of No. 1, No. 5, and No. 9 represented in Figure 8, decreasing by about 25%. While analyzing the surface hardness from the point of view of the grinding depth, it can be obtained from the analysis of the figure that the surface hardness increases with the increase of the grinding depth, and the surface hardness increases by 37% in the areas of No. 4, No. 5, and No. 6, and from the areas of No. 7, No. 8, and No. 9 it can be analyzed that the surface hardness increases by 24%. According to the analysis of test 4 according to Figure 11b, the regularity of the effect of step length and grinding depth on surface hardness is similar to that of test 2. The greatest decrease in surface hardness affected by step length is in areas No.1, No.5, and No.9, with a decrease of 40%. The regularity of the effect of the grinding depth on the surface hardness is not strong. According to the above analysis, it can be obtained that in the roughness Ra below 0.2 μm, the grinding depth of test 2 grinding influenced the surface hardness more than the step length, and the grinding step length of test 4 influenced the surface hardness more than grinding depth. Based on the above analysis, it can be obtained that below the roughness Ra of 0.2 μm, if it is necessary to increase the surface hardness, the grinding depth can be increased, or the step length can be reduced appropriately.

According to the analysis in Figure 11c, it can be obtained that the surface hardness decreases with the increase of linear speed in test 2, such as a 31% decrease in areas No. 1, No. 6, and No. 8, a 42% decrease in areas No. 2, No. 5, and No. 7, and a 35% decrease in areas No. 2, No. 4, and No. 9 in the figure. From Figure 11c, it can be seen that the surface hardness decreases with the increase of feed rates, such as an 8% decrease in areas No. 7, No. 8, and No. 9, and a 13% decrease in areas No. 1, No. 2, and No. 3. According to Figure 11d, the surface hardness in test 4 decreased with the increase of grinding wheel linear speed, 55% in areas No. 1, No. 6, and No. 8, and 9% in areas No. 3, No. 5, and No. 7; according to Figure 11d, the surface hardness increased with the increase of feed rate by 25% at the maximum. According to the above analysis, it can be obtained that below the roughness Ra of 0.2 μm, the grinding wheel linear speed has a greater influence on the surface hardness. If the surface hardness needs to be increased, the grinding wheel linear speed can be reduced appropriately, or the feed rate can be reduced.

### 3.3. Effect of Grinding Parameters and Level on Surface Topography

#### 3.3.1. Roughness Ra 0.4 μm–0.2 μm Surface Topography Analysis

The surface of titanium alloy TC4 was photographed using super depth of field and white light interferometer after repeated grinding by an abrasive belt. The surface of test 1 titanium alloy is shown in Figure 12a,b, and in Figure 12c,d, the pictures are the surface of test 2 titanium alloy. It can be seen from the figures that the surface of test 1 has some defects and produces a lot of grinding scratches as the main feature; for example, the grinding in Figure 12a produced 520 μm long defects and 400 μm scratches, and Figure 12b found a 500 μm long defect and a deeper defect of 200 μm long, and the rest had a lot of scratches. In Figure 12c,d two figures have more scratches, where a 780 μm long scratch exists on Figure 12d, but no defects are present. As shown in Figure 13, for the white light interferogram, it can be seen from Figure 12a that test 1 produced a large gouge with a height difference of 7 μm, while test 2 had a height difference of 1.5 μm and did not produce a large gouge.

#### 3.3.2. Surface Topography Analysis for Roughness Ra below 0.2 μm

As Figure 14 shows the surface morphology with roughness Ra below 0.2 μm, the surface of titanium alloy TC4 of test 2 is mainly characterized by grinding scratches and surface pits according to Figure 14a. As shown in Figure 14a, there are more scratches on the surface of test 2. The more obvious one is about 200 μm long, and there is another surface scratch of 300 μm in length. Figure 14b shows only slight scratches and no surface pits are found. As shown in Figure 15a, test 2 showed deeper gouges, the deepest depth was about 2 μm and the average was about 1 μm, and the textured ground by the abrasive belt could also be seen very clearly on the graph. While the grinding marks shown in Figure 15b are smaller, the deepest depth is about 1 μm, the average is about 0.5 μm, and there is no obvious grinding texture on the graph.

## 4. Discussion

### 4.1. Mechanism of the Influence of Grinding Parameters and Levels on Compression Residual Stresses

According to the analysis in Figure 5, compression residual stresses are more influenced by the step length between roughness Ra 0.4 μm–0.2 μm. When the step length is reduced, the grinding spacing is small, which leads to multiple grindings in the same area, resulting in greater compression residual stresses. Since VSM abrasive belts are made using silicon carbide build-up and 3M abrasive belts are bonded using alumina resin, and 3M abrasive belts are manufactured in a pyramid shape, there is more force during grinding [40,41,42]. The belt diagram is shown in Figure 16. Since the 3M abrasive belt uses alumina particles, the material is harder and squeezes the titanium alloy surface during grinding, producing larger grooves as shown in Figure 13a, thus leaving greater compression residual stresses, as shown in Figure 4. Test 1 and Test 3 (3M abrasive belt) grinding surface compression residual stresses were higher than Test 2 and Test 4 (VSM abrasive belt) grinding. Therefore, the greater compression residual stresses generated by grinding with 3M abrasive belts should be achieved by reducing the abrasive belt linear speed. The VSM abrasive belt uses silicon carbide particles, which is a softer material, and generates greater compression residual stresses by wearing itself during grinding, so a suitable feed rate should be set. According to the analysis in Figure 4, it can be obtained that the 3M belt grinding compression residual stress is not as high as that of VSM belt grinding below a roughness Ra of 0.2 μm. This is because, under high precision conditions, the 3M belt grinding force will be smaller than that between the roughness Ra of 0.4 μm–0.2 μm. The heat generated by the silicon carbide particles is higher than that of aluminum oxide, resulting in a somewhat higher compression residual stress in VSM abrasive belts.

Zhao et al. [43] investigated the residual stresses due to grinding heat from the temperature side and found that the compressive residual stresses increased with the increase of feed rate under the controlled grinding temperature of 500 °C. During grinding, the grinding surface layer generates thermal expansion under the action of grinding heat, while the substrate temperature is low at this time, and the thermal expansion of the grinding surface layer is limited by the substrate and generates compressive stresses.

### 4.2. Mechanism of the Influence of Grinding Parameters and Levels on Surface Hardness

Since VSM abrasive belts are made using silicon carbide build-up and 3M abrasive belts are bonded using alumina resin, and 3M abrasive belts are manufactured in a pyramidal shape, the force during grinding is greater. According to Figure 9, it can be obtained that between roughness Ra 0.2 μm–0.4 μm, compared to test 3, test 4 linear speed and feed rate affect the surface hardness more. Figure 16 shows that the aluminum oxide material used in the 3M abrasive belts is harder, and the phenomenon of wear-off occurs during the grinding process. No surface grinding is performed on the abrasive grain as a whole, but only the pyramidal tops are ground on the surface, thus reducing a large amount of heat, which in turn reduces machine hardening and other conditions. The passivation of the abrasive belt improves surface integrity and thus fatigue life [44]. According to the above analysis, if it is necessary to increase the surface hardness at this roughness Ra, the grinding depth can be reduced, the grinding wheel linear speed can be reduced, or the feed rate can be increased. In addition, below a roughness Ra of 0.2 μm, the temperature is lower due to the high wear during VSM belt grinding, which reduces the process of hardening in belt grinding. If it is necessary to increase the surface hardness, you can increase the grinding depth, reduce the step length, or reduce the grinding wheel’s linear speed.

### 4.3. Mechanism of the Influence of Grinding Parameters and Levels on Surface Topography

As shown in Figure 12, the surface of test 1 and test 3 (using 3M abrasive belts) produced defects, while the surface of test 2 and test 4 (using VSM abrasive belts) did not show defects. This is caused by the different abrasive belts. The 3M abrasive belt is harder, so more scratches appear during the grinding process, and the high temperature and pressure generated by grinding cause the abrasive belt to wear off and further produce defects. Due to the silicon carbide material used in VSM abrasive belts, the processing process is stacked, so the abrasive belts are softer. VSM abrasive belts show some defects during rough grinding (plate roughness Ra of 1.6 μm grinding to the roughness Ra of 0.4 μm) due to the unevenness of the plate, resulting in different forces. However, the number was small, while this did not occur during grinding with the roughness Ra of 0.2 μm or less. Figure 17 is an electron microscope shot of titanium alloy TC4 after polishing using wire cutting, as shown in Figure 17b for the cross-sectional view, from which it can be seen that the grinding layer is about 30 μm, the sub-surface is smoother, there are no large craters, and surface scratches are the main surface special diagnosis. This is consistent with the photograph taken with the white light interferometer. The appearance of craters in Figure 14a can be seen by the two figures in Figure 17c,d as a result of 3M abrasive belt grinding, where the particles are too hard and appear to scratch out the grains and, at the same time, scratch the surface. In Figure 17e, f it can be seen that the surface ground by VSM abrasive belt produced a shallow texture and also did not appear to scratch the surface by 3M abrasive belts. It can be obtained from the figure that better surface topography is needed and the VSM abrasive belt that needs to be used is better.

## 5. Conclusions

We aimed to investigate the effect of grinding parameters and levels on other surface integrity of titanium alloy TC4 under low roughness Ra (0.4 μm–0.2 μm/0.2 μm or less) conditions. This paper investigates the effects of different combinations of parameters and levels on the compressive residual stress, surface hardness, and surface topography of titanium alloys ground by the same type of different types of abrasive belts under the same roughness. Based on the above-mentioned laws, this provides a basis for the engineering improvement of single surface integrity. Several conclusions can be drawn from the experimental results:At a roughness Ra of 0.4 μm–0.2 μm, tests 1 and 3, the compressive residual stress decreases with increasing linear speed and increases with increasing feed rate. The analysis shows that to increase the residual stresses in engineering, the grinding belt’s linear speed can be reduced, and a feed rate of 2 mm/s can be selected. At a roughness Ra of 0.2 μm or less, test 2,4, the compressive residual stress decreases with increasing linear speed and increases with increasing feed rate. The analysis leads to the engineering need to increase the residual stress, which requires reducing the grinding belt’s linear speed and selecting a feed rate of 2 mm/s. Even with different abrasive belts and different roughness, it can be seen from the test results that the residual stress decreases with increasing linear speed and increases with increasing feed rate, which is caused by the large amount of heat generated during the grinding process and thus the cooling of the substrate.Between a roughness Ra of 0.4 μm and 0.2 μm, the surface hardness of test 1 was maximum at a step length of 2 mm; the surface hardness decreased with increasing grinding depth. The surface hardness of test 3 increased with increasing step length and decreased with increasing grinding depth. At a roughness Ra of 0.2 μm or less, the surface hardness of test 2 decreased with the increase of linear speed and increased with the increase of grinding depth; the surface hardness of test 4 decreased with the increase of linear speed and decreased with the increase of step length. The analysis of the test results showed that the surface hardness did not show a good regularity when the roughness Ra was between 0.4 μm and 0.2 μm, while the influence of the linear speed and grinding depth was greater below 0.2 μm. Therefore, if the surface hardness needs to be increased in engineering, the linear speed can be reduced, or the grinding depth can be increased.At a roughness Ra of 0.4 μm or less, the surface topography of VSM belt grinding was better than that of 3M belt grinding, and VSM belt grinding can be chosen for grinding when better surface quality is needed. Using a 3M abrasive belt to grind titanium alloy surfaces produces larger residual stress, but the surface topography is poorer.

## Figures and Tables

**Figure 1 micromachines-13-01950-f001:**
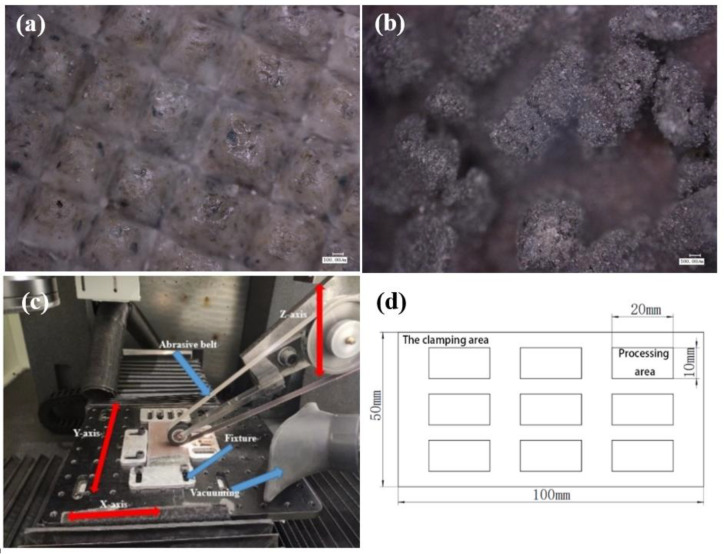
(**a**) for 3M abrasive belts and (**b**) for VSM abrasive belts; (**c**) specimen grinding and (**d**) grinding area.

**Figure 2 micromachines-13-01950-f002:**
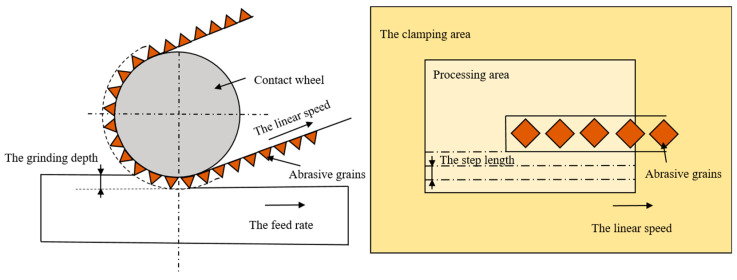
Meaning of grinding parameters.

**Figure 3 micromachines-13-01950-f003:**
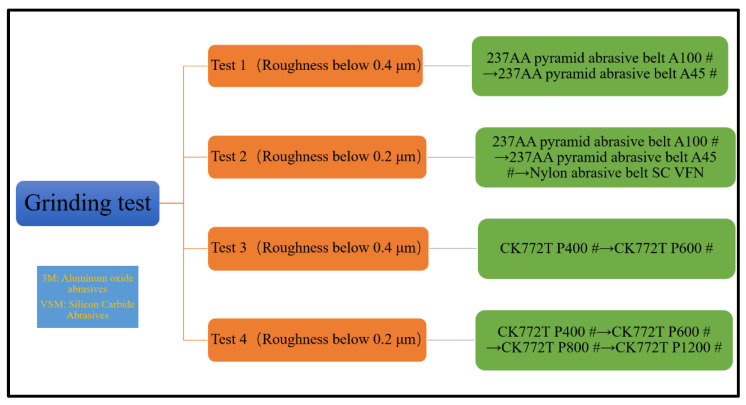
Grinding test abrasive belt sequence flow chart.

**Figure 4 micromachines-13-01950-f004:**
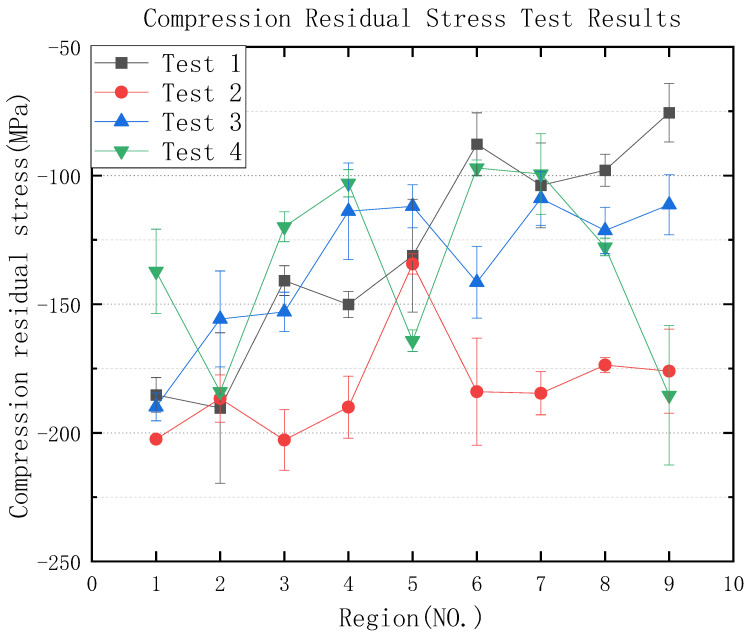
Compression residual stress test results.

**Figure 5 micromachines-13-01950-f005:**
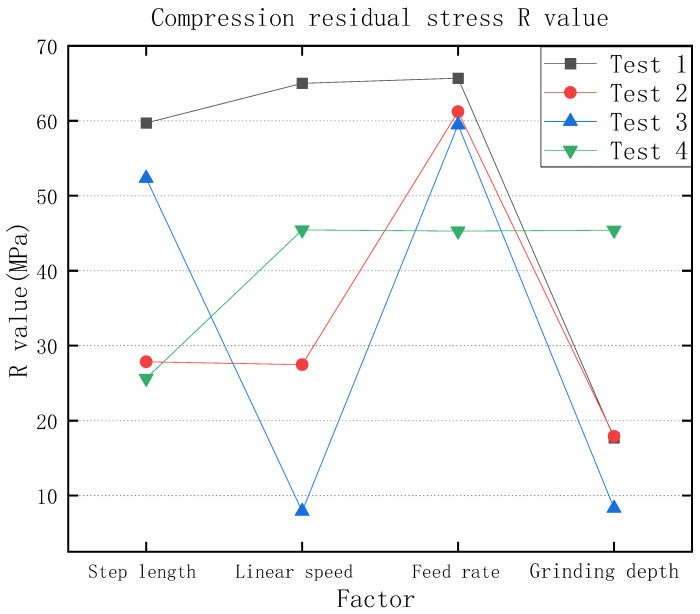
Compression residual stresses by extreme difference analysis.

**Figure 6 micromachines-13-01950-f006:**
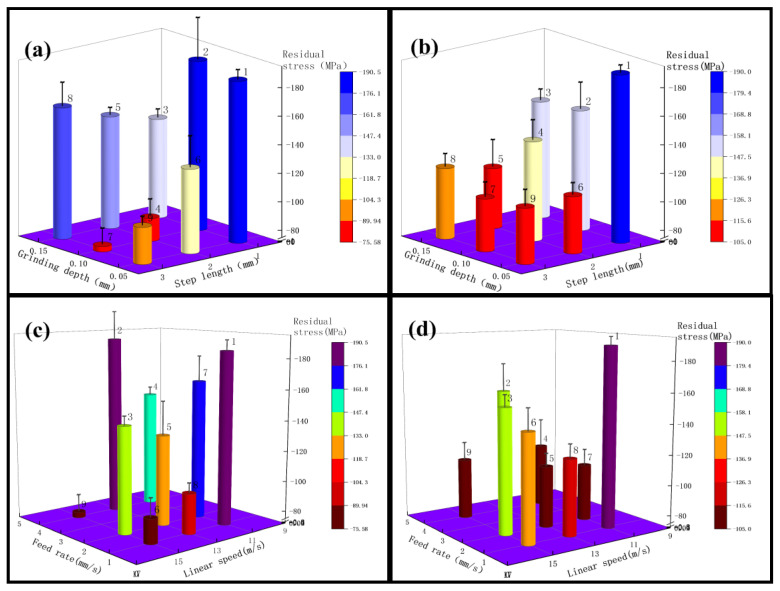
(**a**) for test 1, variables are grinding depth and step length, (**b**) for test 3, variables are grinding depth and step length; (**c**) for test 1, variables are linear speed and feed rate, (**d**) for test 3, variables linear speed and feed rate.

**Figure 7 micromachines-13-01950-f007:**
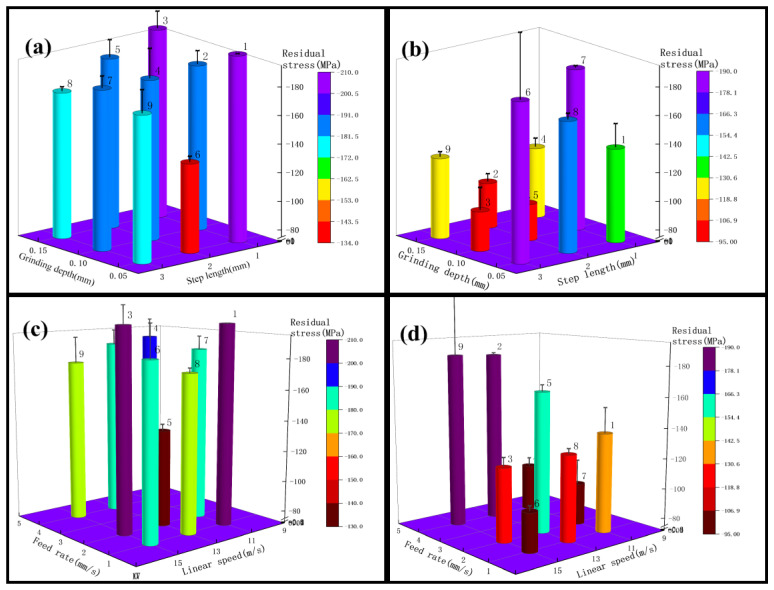
(**a**) for test 2, variables are grinding depth and step length, (**b**) for test 4, variables are grinding depth and step length; (**c**) for test 2, variables are linear speed and feed rate, (**d**) for test 4, variables linear speed and feed rate.

**Figure 8 micromachines-13-01950-f008:**
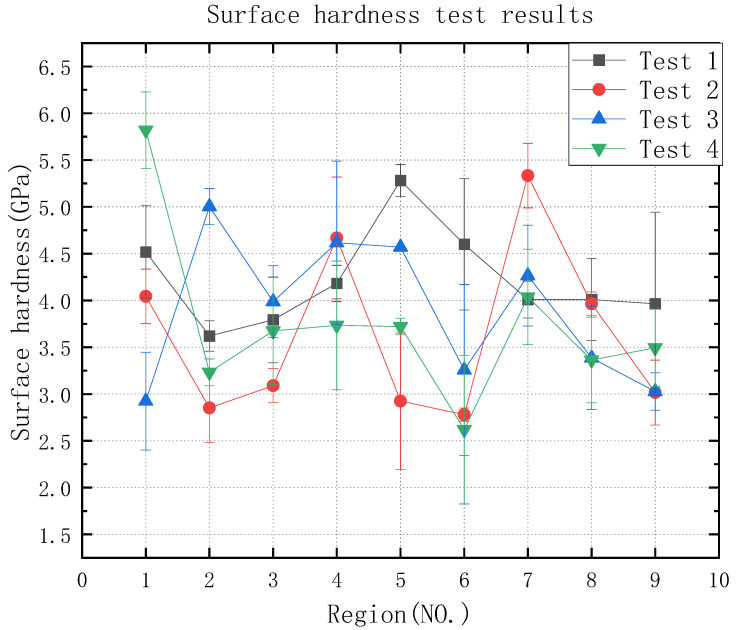
Graph of surface hardness test results.

**Figure 9 micromachines-13-01950-f009:**
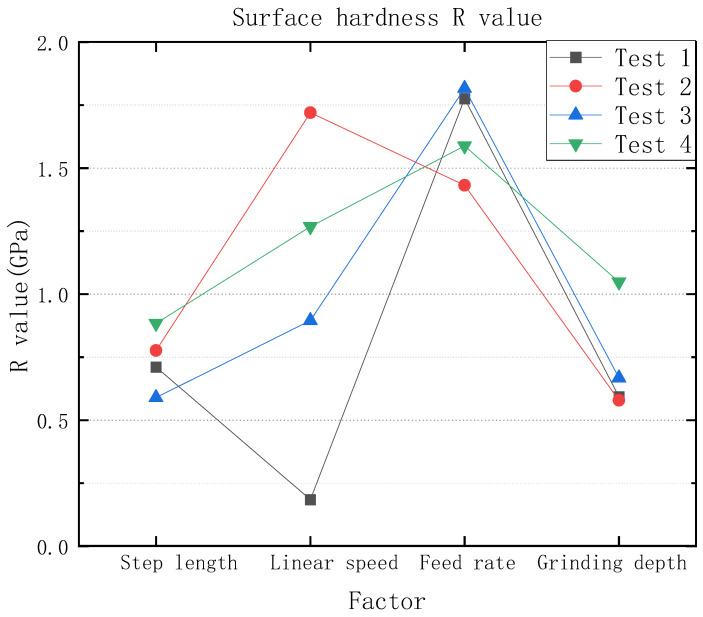
Surface hardness graph for polarization analysis.

**Figure 10 micromachines-13-01950-f010:**
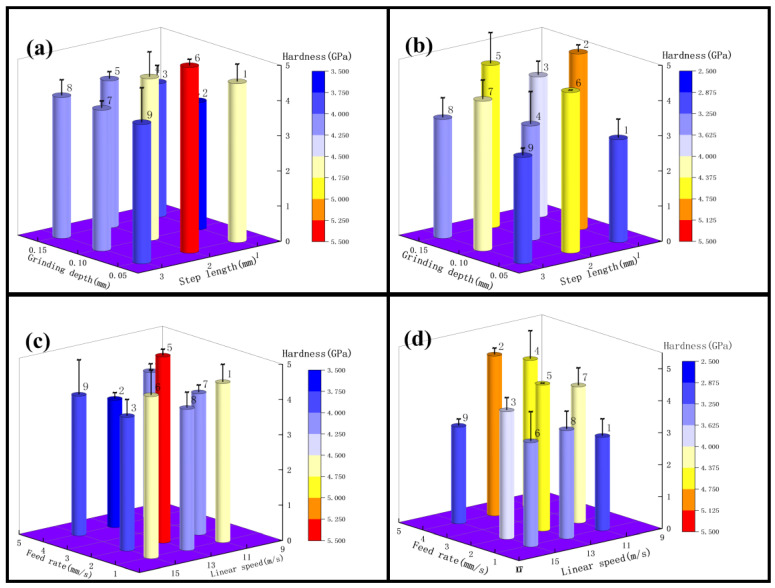
(**a**) for test 1, variables are grinding depth and step length; (**b**) for test 3, variables are grinding depth and step length; (**c**) for test 1, variables are linear speed and feed rate; (**d**) for test 3, variables linear speed and feed rate.

**Figure 11 micromachines-13-01950-f011:**
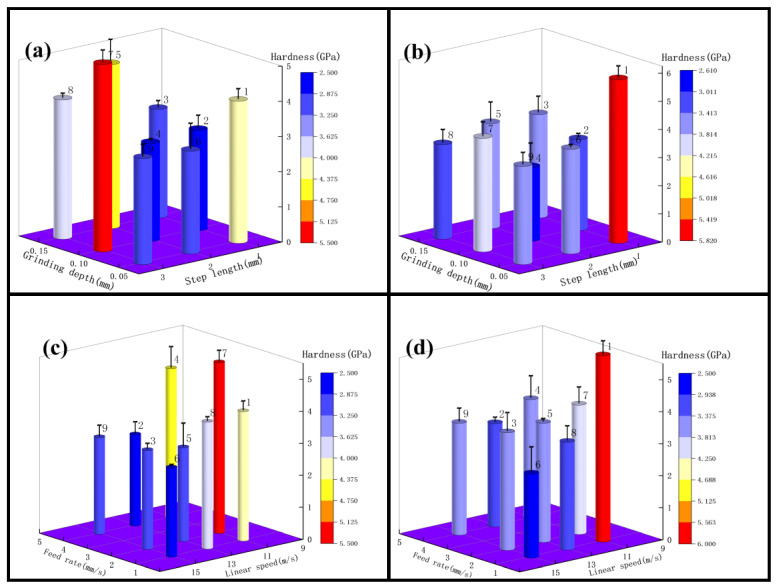
(**a**) For test 2, the variables are grinding depth and step length; (**b**) for test 4, the variables are grinding depth and step length; (**c**) for test 2, the variables are linear speed and feed rate; (**d**) for test 4, the variables are linear speed and feed rate.

**Figure 12 micromachines-13-01950-f012:**
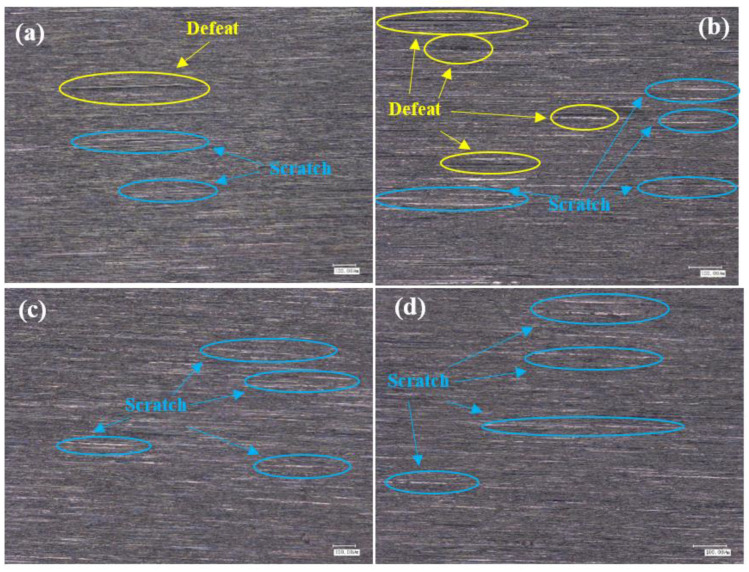
(**a**,**b**) Shows the surface of test 1, and (**c**,**d**) shows the surface of test 2.

**Figure 13 micromachines-13-01950-f013:**
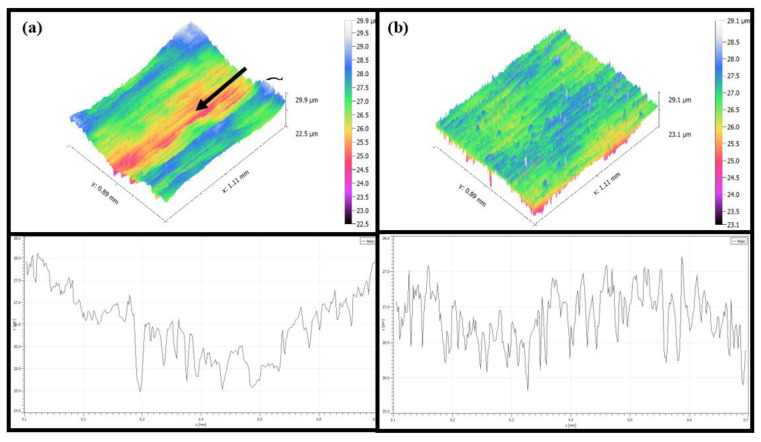
(**a**) Shows the surface of test 1, and (**b**) shows the surface of test 3.

**Figure 14 micromachines-13-01950-f014:**
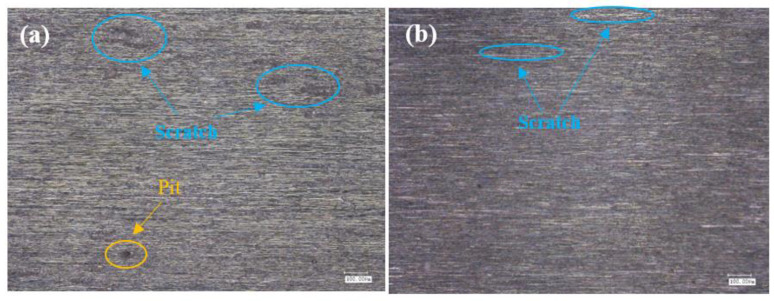
(**a**) The surface of test 1, (**b**) the surface of test 2.

**Figure 15 micromachines-13-01950-f015:**
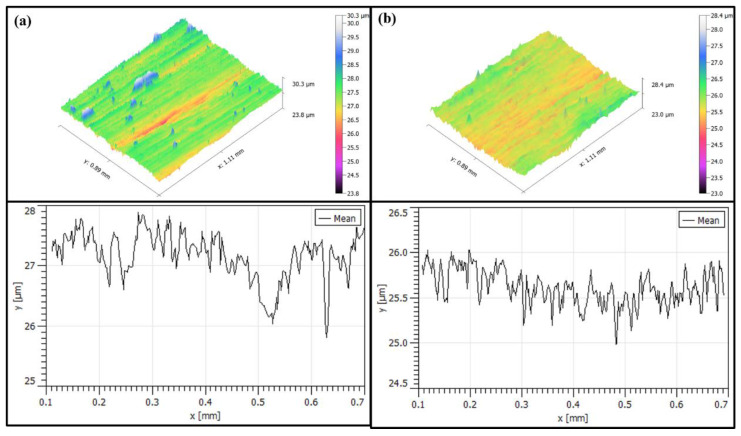
(**a**) Shows the surface of test 2, (**b**) shows the surface of test 4.

**Figure 16 micromachines-13-01950-f016:**
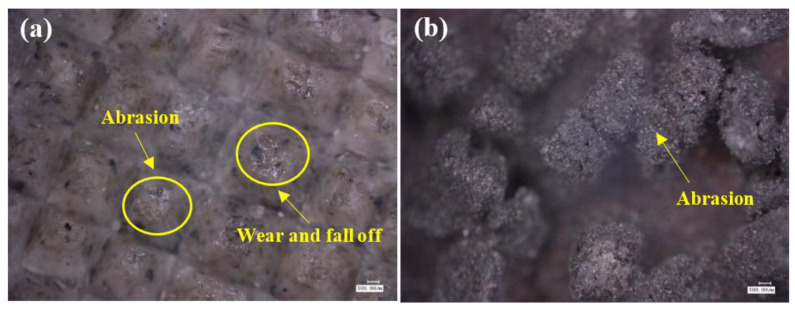
(**a**) shows 3M abrasive belt grit, (**b**) shows VSM abrasive belt grit.

**Figure 17 micromachines-13-01950-f017:**
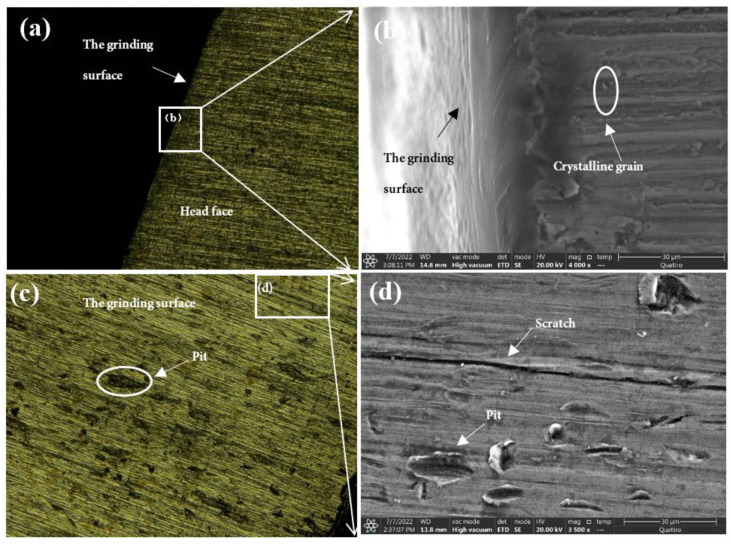
(**a**,**c**,**e**) are super depth-of-field shots, (**b**,**d**,**f**) are electron microscope shots; (**a**,**b**) are end surfaces, (**c**–**f**) are ground surfaces.

**Table 1 micromachines-13-01950-t001:** Approximate chemical composition of titanium alloy TC4 (wt%).

Element	Al	V	Fe	O	C	N	H	Ti
wt%	6.12	3.95	0.12	0.085	0.01	0.005	0.004	Bal

**Table 2 micromachines-13-01950-t002:** Mechanical properties of TC4.

Materials	Strength (MPa)	Compressive Strength (MPa)	Elastic Modulus (GPa)	Hardness (HRC)
TC4	1100	895	110	30

**Table 3 micromachines-13-01950-t003:** Orthogonal test parameters.

NO.	Step Length (mm)	Linear Speed (m/s)	Feed Rate (mm/s)	Grinding Depth (mm)
1	1	11	1	0.05
2	1	13	4	0.10
3	1	15	2	0.15
4	2	11	4	0.15
5	2	13	2	0.05
6	2	15	1	0.10
7	3	11	2	0.10
8	3	13	1	0.15
9	3	15	4	0.05

**Table 4 micromachines-13-01950-t004:** Surface roughness experimental results.

NO.	Text 1	Text 2	Text 3	Text 4
1	0.219	0.152	0.288	0.154
2	0.338	0.161	0.247	0.141
3	0.319	0.160	0.267	0.195
4	0.239	0.197	0.320	0.189
5	0.307	0.180	0.283	0.171
6	0.302	0.160	0.278	0.176
7	0.220	0.138	0.299	0.184
8	0.300	0.150	0.277	0.182
9	0.211	0.192	0.277	0.159

**Table 5 micromachines-13-01950-t005:** Compression residual stress test results.

NO.	Text 1	Text 2	Text 3	Text 4
1	−185.26	−202.45	−189.90	−137.28
2	−190.36	−186.66	−155.75	−183.99
3	−140.85	−202.80	−153.02	−119.90
4	−150.11	−190.01	−113.90	−103.00
5	−131.16	−134.37	−111.95	−164.20
6	−87.80	−184.02	−141.49	−97.08
7	−163.82	−184.63	−108.94	−99.41
8	−97.93	−173.68	−121.36	−127.77
9	−75.58	−176.02	−111.35	−185.42

**Table 6 micromachines-13-01950-t006:** Graph of surface hardness test results.

NO.	Text 1	Text 2	Text 3	Text 4
1	4.5185	4.0430	2.9239	5.8187
2	3.6199	2.8515	5.0026	3.2323
3	3.7923	3.0896	3.9876	3.6743
4	4.1818	4.6676	4.6170	3.7334
5	5.2813	2.9247	4.5702	3.7210
6	4.5999	2.7800	3.2572	2.6190
7	4.0087	5.3344	4.2639	4.0380
8	4.0101	3.9638	3.3852	3.3634
9	3.9644	3.0158	3.0266	3.4936

## Data Availability

Not applicable.

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
