# Peer review of "Study of Surface Integrity of Titanium Alloy (TC4) by Belt Grinding to Achieve the Same Surface Roughness Range"

_micromachines, 2022, doi:10.3390/mi13111950_

Round 1

Reviewer 1 Report

Surface integrity of titanium alloy (TC4) by belt grinding was researched through repeated surface grinding of titanium alloy TC4 in this manuscript. It would be interested to researchers in engineering field.

Some issues should be addressed for the following review process.

1. The abstract should have a deeply revision, some words are very inappropriate, for example, 'modern performance' in line 12, 'achieve' in  line 13, 'material' in line 15 , 'there was a problem' in line 18 and 19, 'thesis' in line 21, and so on. Also, the sentence 'the residual stress increased by 48% with feed rate and the surface 20 hardness decreased by 55% with linear speed' make me confused, maybe some information was missed.

2. What's the original contribution of this research compared with existed researches? please comment.

3. The table 1 and 2 are well known from books and references.

4. More grinding parameters should be supplymented in Figure 2, I think it would be easy reading for readers.

5. The units in most figures are missing. pls revise.

6. The conclusion section should be rewritten, it cannot be a repeated description of the experiment results. 

Reviewer 2 Report

Comments are presented in the attached file.

Reviewer 3 Report

1. The author should explain in detail why roughness is divided into high precision (0.4um to 0.2um roughness) and ultra-high precision (below 0.2um roughness), which is not very detailed in the introduction. This is critical because it is the basis of this article.

2. The effects of grinding parameters and levels on residual stress, hardness and surface morphology under Taguchi experimental conditions are introduced in detail. The experimental data are detailed and true, but the content is too much. On the contrary, the depth of the analysis in the discussion part of the experimental results is not enough. I hope the author can make some supplements in terms of the influence mechanism.

3. For the classical belt grinding literature, the author should quote in the introduction and explain what improvements or changes have been made in this paper.

Such as:

[1] X. Ren, B. Kuhlenkötter, Real-time simulation and visualization of robotic belt grinding processes, Int. J. Adv. Manuf. Technol. 35 (2008) 1090–1099.

[2] G. Xiao, Y. Huang, Constant-load adaptive belt polishing of the weak-rigidity blisk blade, Int. J. Adv. Manuf. Technol. 78 (2015) 1473–1484.

[83] C. Qu, Y. Lv, Z. Yang, X. Xu, D. Zhu, S. Yan, An improved chip-thickness model for surface roughness prediction in robotic belt grinding considering the elastic state at contact wheel-workpiece interface, Int. J. Adv. Manuf. Technol. 104 (2019)3209–3217.

[4] S. Wu, K. Kazerounian, Z. Gan, Y. Sun, A simulation platform for optimal selection of robotic belt grinding system parameters, Int. J. Adv. Manuf. Technol. 64 (2013) 447–458.

[5] D. Axinte, M. Kritmanorot, M. Axinte, N. Gindy, Investigations on belt polishing of heat-resistant titanium alloys, J. Mater. Process. Technol. 166 (2005) 398–404.

[6] E. Brinksmeier, J. Aurich, E. Govekar, C. Heinzel, H.-W. Hoffmeister, F. Klocke, J. Peters, R. Rentsch, D. Stephenson, E. Uhlmann, Advances in modeling and simulation of grinding processes, CIRP Ann. 55 (2006) 667–696.

Round 2

Reviewer 1 Report

The revised version is better. However, the term 'thesis' in the abstract hasn't been amended.

Reviewer 2 Report

I appreciate the authors' efforts in making improvements. However, the current version still contains several issues that require attention:

1.       The authors still use the generic phrase roughness when referring to the specific roughness parameter Ra. This needs to be clarified, e.g. in the abstract.

2.       The authors still have not fully presented the kinematics of the process and have not shown in the diagram/drawing the grinding parameters that were the input variables in the experiment.

3.       The parameters for measuring roughness, topography, hardness, and residual stress are still not provided. Thus, there is no way to verify the correctness of the measurements.

4.       It is still not explained what the” Impact” and “R values” are in Figures 4 and 8.

5.       The description of the results relating to Figures 5, 6, 9, and 10 refer to “No.” - however, there are no such designations in the figures and tracing the correctness of the text for the reader is still very difficult. It is sufficient to add the “No.” numbers next to the bars in question (in any graphic editor).

6.       To make it easier to follow the text, please add in figure 2 which materials are VSM and 3M.

7.        The bar charts are missing units. 

Reviewer 3 Report

It can be accepted and published with minor English language spell check required.

Round 3

Reviewer 2 Report

The authors have taken most of the comments into account. Unfortunately, they did not address the most important comment concerning the methodology of the hardness, roughness/topography, and residual stress measurements carried out.

The parameters for measuring roughness, topography, hardness, and residual stress are still not provided. Thus, there is no way to verify the correctness of the measurements.

Round 4

Reviewer 2 Report

I have no further comments.